# Building Luganda Machine Translation models for the Medical Domain

## Abstract

Globalization and migration have highlighted the critical need for effective cross-language communication, particularly in healthcare. In Uganda, a multilingual nation where Luganda is widely spoken, language barriers in predominantly English-speaking medical settings often lead to misunderstandings, misdiagnoses, and compromised patient care. This study aims to mitigate these issues by developing a machine translation model built for medical communication, specifically targeting translations from English to Luganda within the context of malaria diagnosis and community engagement. Utilizing recent advancements in machine learning, this research involves curating a parallel medical corpus, training a transformer-based model with domain-specific adapters, and rigorously evaluating the model's accuracy and cultural sensitivity. The results demonstrate that the `MarianMT-Adapter LoRa` model, when combined with active learning, achieved a significant improvement in translation quality, evidenced by a BLEU score increase to **56%**. This model effectively reduced translation errors and preserved the contextual integrity of medical texts. The findings are anticipated to enhance healthcare communication, reduce disparities, and improve access to medical knowledge for Luganda-speaking communities, providing a blueprint for similar efforts in other multilingual environments.

## 1 Introduction

In today's rapidly digitalising world, the need for effective communication across different languages is more critical than ever, particularly in essential sectors such as healthcare. Language barriers in medical settings can lead to serious consequences, including misunderstandings, misdiagnoses, and inadequate treatment, all of which compromise patient care and safety Mustafa et al. (2023). This issue is especially relevant in multilingual societies like Uganda, where linguistic diversity often clashes with the predominantly English-speaking medical workforce Hull (2016). Among Uganda's many languages, Luganda is widely spoken in the central region, including the capital, Kampala Ssempuuma (2019). Despite its prevalence, there exists a substantial language gap in medical settings where English remains the primary mode of communication. This gap hinders effective patient-provider interactions and limits access to essential healthcare services for Luganda-speaking communities.

This problem is aggravated by a lack of skilled medical translators, especially in rural and under-resourced areas, and the inefficiencies of current human-mediated translation services, which often fail to account for cultural nuances and the complexity of medical terminology. Artificial Intelligence (AI), particularly in the form of Machine Translation (MT), offers a promising solution. Advances in AI and Natural Language Processing (NLP) have led to significant improvements in translation accuracy, but these developments have primarily focused on widely spoken languages Khasawneh & Al-Amrat (2023). Lesser-known languages like Luganda remain underrepresented, especially in specialized fields such as healthcare.

The scarcity of parallel text corpora in low-resource languages like Luganda presents a challenge for traditional, supervised MT models. However, recent developments in unsupervised MT techniques, which do not rely on parallel texts, offer a potential path forward. In the context of healthcare, particularly in the fight against malaria—a disease with high prevalence in Uganda—there is an urgent need to communicate accurate information about symptoms, prevention, and treatment in

local languages Chemonges Wanyama et al. (2021). This study aims to leverage AI technologies, specifically unsupervised MT models, to develop a tool that can accurately translate medical texts related to malaria from English to Luganda. By addressing the language barrier in medical communication, this research seeks to improve healthcare outcomes in Luganda-speaking communities, making healthcare more inclusive and accessible. The development of this MT model not only offers a technological solution but also serves as a public health intervention, potentially transforming healthcare delivery in linguistically diverse regions.

## 2 LITERATURE REVIEW

This section reviews existing literature related to supervised, semi-supervised, and unsupervised learning techniques, focusing on their applications to machine translation and healthcare. Machine learning (ML) has been applied to numerous healthcare tasks, including diagnosis, treatment, drug discovery, and healthcare administration, while also advancing machine translation (MT) for low-resource languages.

Supervised learning, where labeled data is provided to train models, has shown significant success in domains with abundant labeled datasets. In the context of machine translation and healthcare, supervised models, such as Neural Machine Translation (NMT), rely on parallel corpora to achieve high performance Abrishami et al. (2020). In healthcare, supervised learning techniques have improved diagnostic accuracy and personalized treatment by processing patient data Ahmad et al. (2021); Ballester & Carmona (2021).

Unsupervised learning, which operates without labeled data, has become essential for low-resource languages where labeled corpora are scarce. Methods like Unsupervised NMT (UNMT) utilize monolingual data and techniques such as back-translation to bridge language barriers, especially in low-resource settings Artetxe et al. (2017); Lample (2019). This is particularly relevant to healthcare, where language diversity complicates communication and access to medical resources Haddow et al. (2021).

Despite these advancements, several research gaps persist. Supervised learning models struggle with scalability in low-resource environments due to their reliance on extensive labeled data. Moreover, unsupervised approaches face technical challenges in maintaining translation accuracy, particularly in domain-specific contexts like healthcare, where precise terminology is critical Conneau & Lample (2019). Additionally, the lack of robust parallel corpora for low-resource languages in healthcare poses a significant barrier to developing accurate MT systems for medical content Artetxe et al. (2017).

To address these gaps, this research focuses on improving unsupervised models for medical translation in low-resource settings. By leveraging monolingual data and reinforcement learning techniques, we aim to enhance translation quality and overcome existing limitations. Applying contextual annotation frameworks, as used in MQM Utiyama & Wang (2023), can further refine error detection in medical translations, contributing to more reliable models.

## 3 METHODOLOGY

This section introduces the methodological concepts used in our experiments to achieve the objectives of this research. It encompasses the processes of data collection, model training, and the application of various techniques to optimize machine translation models tailored for the medical domain, specifically focusing on translating content from English to Luganda.

### 3.1 DATASET DESCRIPTION

The dataset utilized in this study was sourced from a variety of collections, with the primary aim of addressing the gap in medical translations between English and Luganda. Several factors were considered during the selection of these datasets, including relevance to the medical domain, data quality, and scale, as well as the potential to support the development of an effective machine translation model. The dataset is rich in medical terminologies and phrases, which are crucial for ensuring accurate and reliable translation.

## 3.2 DATA COLLECTION

The primary focus of this study was the creation of an English-Luganda parallel corpus, specifically for the medical domain. Since Luganda was not originally part of the UFAL Medical Corpus Rasheed et al. (2021), the strategy employed involved back translation using Google Translate. This method translated English medical texts into Luganda, creating a collection of parallel sentences relevant to the medical domain in both languages.

### 3.2.1 BASELINE PARALLEL CORPUS

To establish a baseline for machine translation, the English-Luganda dataset was used for model training. To establish a robust baseline for machine translation, an English-Luganda dataset comprising 81,050 sentences was used for model training. This dataset was compiled from several sources, which include: Makerere Parallel Corpus, Gendered Parallel Corpus and Sunbird AI Parallel Corpus SALT datasetAkera et al. (2022).

### 3.2.2 MEDICAL CORPUS AND BACK TRANSLATION

The UFAL Medical Corpus was expanded to include Luganda through back translation. This corpus contains 155,193 sentences from various medical specialties, providing comprehensive coverage of topics within the medical field. The corpus is organized by specialties such as General Medicine, Surgery, Radiology, and Neurology, among others, ensuring that the parallel sentences generated through back translation cover a broad range of medical concepts.

In total, the combined dataset contains 3,603,767 sentences, with an English token count of 32,941,227 and a Luganda token count of 29,173,358. Figure 1 illustrates the distribution of medical topics within the corpus.

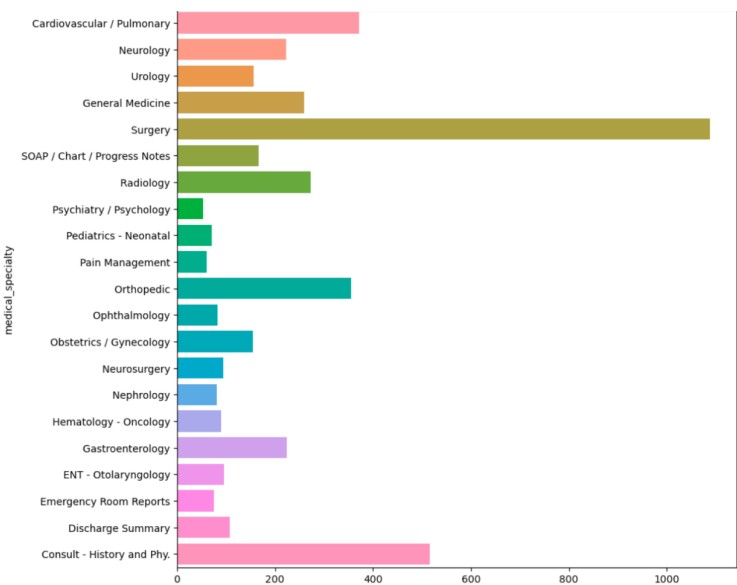

Figure 1: Topic Distribution of UFAL Medical Corpus

### 3.2.3 EVALUATION CORPUS

An evaluation corpus focused on malaria and community engagement was specifically developed to test the translation models in real-world medical scenarios. This evaluation corpus consists of 500 parallel sentences, centered on the domain of malaria, providing a relevant and practical test set to assess the model's performance.

| English | Type | Luganda |
|---------|------|---------|
| Symptoms of malaria include fever, headache, and vomiting. | Symptoms | Obubonero bw'omusujja gw'ensiri mulimu omusujja, okulumwa omutwe n'okusesema. |
| Malaria can be diagnosed through blood tests to detect the parasite. | Treatment | Omusujja gw'ensiri gusobola kulabibwa na kuggya musaayi ku muntu era ne gukeberebwa. |

Table 1: Sample Sentences from the Malaria Evaluation Corpus

### 3.2.4 NO LANGUAGE LEFT BEHIND (NLLB) DATASET

In addition to the manually collected data, several open-source corpora were incorporated into the parallel corpus. These include resources from Wikimedia, CCAligned, GNOME, tico-19, XLEnt, QED, and Ubuntu, among others, as shown in Table 2.

| Corpus | Sentences | Tokens |
|--------|-----------|--------|
| NLLB | 3,560,449 | 32,941,227 |
| Wikimedia | 19,170 | 176,544 |
| CCAligned | 14,702 | 135,568 |
| GNOME | 4,578 | 42,209 |
| tico-19 | 3,071 | 28,310 |
| XLEnt | 1,054 | 9,713 |
| QED | 740 | 6,811 |
| Tatoeba | 3 | 27 |
| Ubuntu | 3,603,767 | 29,173,358 |
| **Total** | 7,207,534 | 62,513,767 |

Table 2: Resources for the English-Luganda Corpus

### 3.3 DATASET PREPARATION

Before training the models, the dataset underwent a meticulous preparation process. This involved data cleaning to remove inconsistencies, text normalization to standardize variations in spelling and format, and tokenization to break down sentences into smaller, manageable units. Back translation techniques were employed to augment the dataset, providing greater robustness and variety for the model.

The data was split into three sets to facilitate comprehensive model training and evaluation: the training set consists of 56,734 sentences, the testing set contains 12,159 sentences, and the validation set holds 12,157 sentences. This distribution ensures a balanced evaluation of the model's performance.

### 3.4 DATA PREPROCESSING

The newly generated Luganda-English parallel corpus was subjected to a series of preprocessing steps to ensure quality and usability. The steps included:

- **Cleaning:** Removal of noise, errors, and inconsistencies that may have been introduced during data collection and back translation.

- **Normalization:** Standardization of text to account for variations in spelling, grammar, and formatting.

- **Tokenization:** Breaking sentences into smaller units (tokens) to facilitate easier processing by machine learning models.

- **Segmentation:** Dividing the dataset into manageable subsets for training, validation, and testing purposes.

- **Augmentation:** Employing back translation and other techniques to expand the dataset and enhance its diversity and robustness.

These preprocessing steps ensured that the dataset was clean, standardized, and ready for model training. The malaria evaluation corpus, consisting of 500 parallel sentences, was specifically tailored to evaluate the translation models' performance in critical healthcare domains.

## 3.5 MODEL TRAINING

For this study, we employed MarianMT as our baseline machine translation (MT) model Junczys-Dowmunt et al. (2018). MarianMT is a highly efficient and flexible neural MT framework built on the transformer architecture Vaswani (2017). The transformer has become the de facto standard for state-of-the-art MT systems due to its ability to model long-range dependencies and its parallelization capabilities.

The MarianMT model architecture consists of an encoder-decoder structure, both of which are composed of multiple layers of self-attention mechanisms and feed-forward neural networks. The encoder processes the input sentence from the source language, while the decoder generates the corresponding translation in the target language. The self-attention mechanism enables the model to capture contextual dependencies effectively by weighing the importance of various words in the input.

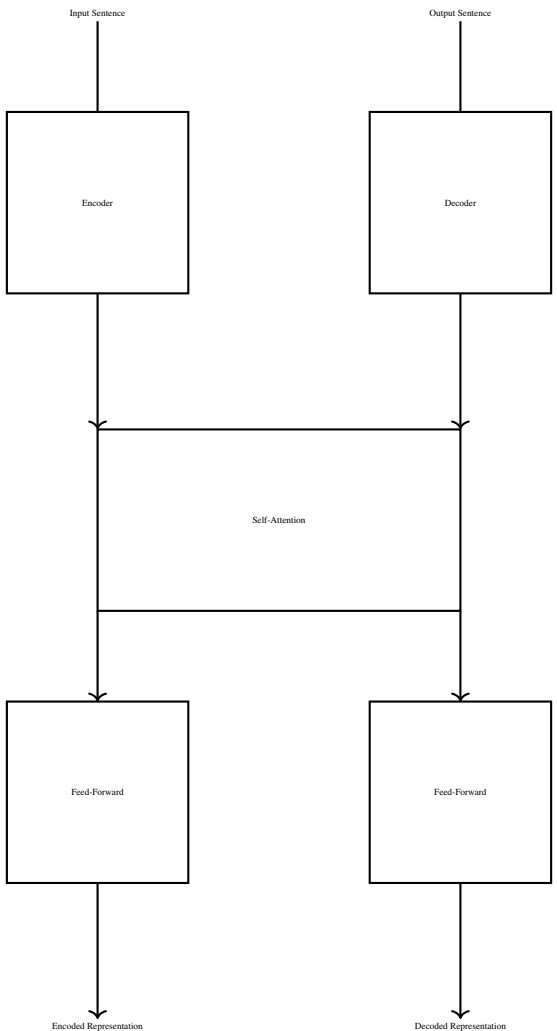

Figure 2: Architecture of the Baseline MarianMT Model

For this work, we adopted the Sunbird AI MarianMT model Akera et al. (2022) as the baseline, trained for `30 epochs` using the Makerere Corpus. This provided a strong foundation for subsequent enhancements.

### 3.5.1 ADAPTER OPTIMIZATION

To improve the performance of the baseline MarianMT model, we incorporated adapter modules using Low-Rank Adaptation (LoRA) Hu et al. (2021). Adapters are lightweight modules added to each layer of the pre-trained model, allowing it to adapt to new tasks or domains with minimal changes to the original parameters. This approach is particularly beneficial when dealing with limited training data or aiming to preserve the generalization capabilities of the pre-trained model.

The adapter modules, while being trained alongside the main model, consist of a much smaller number of parameters, allowing efficient fine-tuning without requiring significant computational resources. In this case, we fine-tuned the MarianMT model with adapters on the NLLB and UFAL datasets, both with and without active learning.

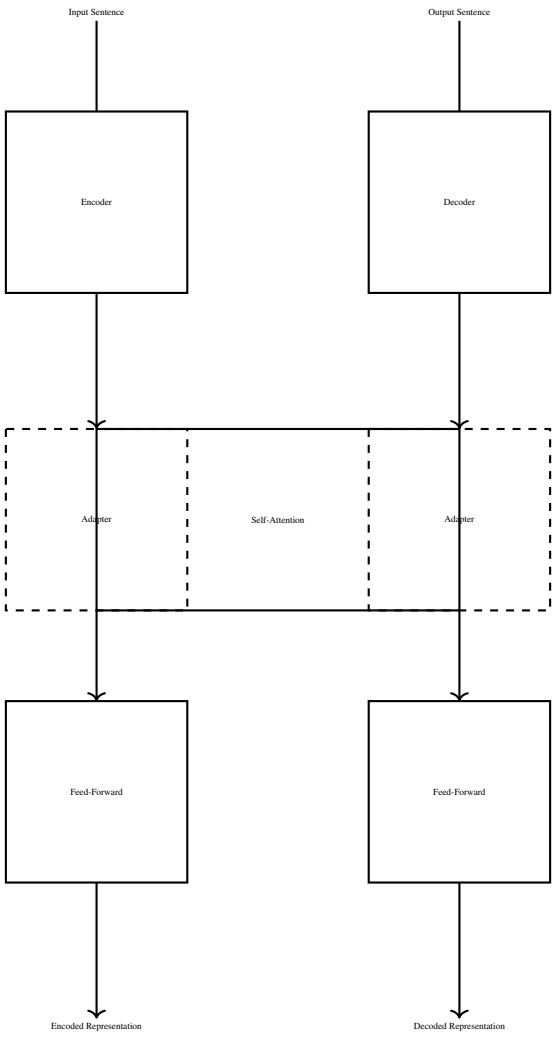

Figure 3: Architecture of the MarianMT Model with Adapter Optimization

The MarianMT-Adapter LoRA model was fine-tuned for `5 epochs` on the NLLB and UFAL datasets, showing promising improvements but requiring further tuning to reach peak performance.

### 3.5.2 ACTIVE LEARNING FOR MODEL IMPROVEMENT

To further enhance our machine translation model, we employed an active learning approach, which iteratively selects the most informative samples from a pool of unlabeled data. This allows the model

to improve its performance more efficiently by focusing on the most challenging data points Zhao et al. (2020). For this study, we utilized uncertainty sampling.

We began by training the model on an initial dataset of 1,000 samples. After the training phase, the model's performance was evaluated using a validation set. Based on this, uncertainty scores were computed from the prediction probabilities using entropy. Samples with the highest uncertainty were then selected for additional training. This process was repeated for five iterations, each time adding 500 new samples, followed by retraining and re-evaluation.

For the implementation, entropy-based uncertainty sampling was employed, selecting samples with higher entropy for retraining. The model was trained using a batch size of 32, leveraging gradient accumulation and mixed precision to optimize memory usage. Evaluation was conducted using BLEU scores on the validation set to assess translation quality.

The MarianMT-Adapter LoRA model, combined with active learning, was subsequently trained for two additional epochs on the malaria corpus, focusing on the most informative samples to improve translation accuracy.

Overall, the training schedule involved the following configurations: MarianMT baseline for `30 epochs`, MarianMT-Adapter LoRA for `5 epochs`, and MarianMT-Adapter LoRA with Active Learning for `2 epochs`. All models were evaluated on the malaria corpus to ensure their relevance to the target medical domain.

## 4 MODEL EVALUATION

### 4.1 EVALUATION METRICS

The performance of our machine translation models was evaluated using both automatic and human assessment metrics. We employed BLEU scores for automatic evaluation and the Multidimensional Quality Metrics (MQM) framework for human evaluation, specifically tailored to the malaria corpus.

#### 4.1.1 BLEU SCORE

The Bilingual Evaluation Understudy (BLEU) score is a widely used automatic metric that compares machine-generated translations to reference translations by calculating the precision of n-grams. A brevity penalty is applied to penalize overly short translations. BLEU scores range from 0 to 1, with higher scores

## 5 RESULTS AND DISCUSSION

### 5.1 RESULTS

This section presents the outcomes of our machine translation models, comparing the performance of various models based on BLEU scores and qualitative analysis. Tables and figures are used to illustrate the translation quality and demonstrate the impact of different optimization techniques.

As seen in Table 3, the baseline MarianMT model, trained on the Makerere Corpus, achieved a BLEU score of **55%**. This model serves as the foundation for further model improvements. However, when the MarianMT model was trained on the NLLB Dataset, its BLEU score slightly dropped to **53%**, reflecting the differences in dataset characteristics.

Introducing adapter optimization with LoRA showed mixed results. The MarianMT-Adapter LoRA model, trained on the NLLB Dataset, experienced a drop in performance with a BLEU score of **50%**. This indicates that while LoRA adapters are effective for certain tasks, they may require further fine-tuning, especially when applied to domain-specific datasets. When the same model was trained on the UFAL Dataset, the BLEU score remained steady at **50%**, suggesting that the dataset provided a stable testing ground for the adapter technique.

A notable improvement was observed when active learning was applied to the MarianMT-Adapter LoRA model. The model achieved a BLEU score of **56%** on the UFAL Dataset, marking a **6%** increase. This demonstrates the effectiveness of active learning in improving model performance by

iteratively selecting the most informative samples for training. Active learning significantly reduced translation errors and enhanced the model's overall translation quality.

| Model Name | BLEU Score (%) | Dataset |
|---|---|---|
| MarianMT Baseline | 55 | Makerere Corpus |
| MarianMT | $53^{-2}$ | NLLB Dataset |
| MarianMT-Adapter LoRa | $50^{-3}$ | NLLB Dataset |
| MarianMT-Adapter LoRa | $50^{0}$ | UFAL Dataset |
| MarianMT-Adapter LoRa with Active Learning | $56^{+6}$ | UFAL Dataset |
| Facebook/NLLB-200-distilled-600M | 27 | N/A |
| Google Translate | 52 | N/A |
| Sunbird/sunbird-en-mul | 53 | N/A |

Table 3: Comparison of MarianMT Models

Qualitative analysis, shown in Tables 4, further emphasizes these findings. For instance, the baseline MarianMT model struggled with translating complex medical terminology, such as "insecticide-treated bed nets." In contrast, the MarianMT-Adapter LoRa with Active Learning significantly improved the translation quality, accurately capturing the meaning of key medical terms. The comparison between baseline and optimized model translations reveals that active learning helps minimize translation errors, making the model more reliable for medical applications.

| Input | Output | Reference |
|---|---|---|
| Symptoms of malaria include fever, headache, and vomiting. | Obubonero bw'omusujja gw'ensiri mulimu omusujja, omutwe n'okusesema. | Obubonero bw'omusujja gw'ensiri mulimu okulumwa omutwe n'okusesema. |
| Prevention methods include insecticide-treated bed nets and anti-malarial drugs. | Enkola z'okutangira mulimu obutimba bw'ebiwuka obulimu ennyunyunta obuwuka n'eddagala ery'okulwanyisa obulwadde. | Enkola z'okutangira omusujja gw'ensiri mulimu okusula mu butimba bw'esiri obunnyikiddwa mu ddagala n'okukozesa eddagala eritangira omusujja gw'ensiri. |

Table 4: Sample translations from the baseline model, highlighting omissions and mistranslations.

## 5.2 DISCUSSION

The results demonstrate both the strengths and limitations of the different models. The MarianMT Baseline model, trained on the Makerere Corpus, provided a strong foundation but showed some limitations in handling specific datasets, as evidenced by the BLEU score drop on the NLLB Dataset.

Adapter optimization with LoRA, though effective for adding task-specific functionality, introduced additional challenges. The decline in performance across datasets suggests that this technique needs further refinement and larger amounts of domain-specific data to achieve the desired results. However, the real breakthrough came when active learning was introduced.

Active learning greatly improved the MarianMT-Adapter LoRa model's performance, as demonstrated by the significant **6%** increase in BLEU score. This approach allowed the model to iteratively focus on the most informative and challenging samples, reducing errors and producing more accurate translations. Figure 4 showcases the improvements in translation quality, where the active learning model corrected many of the errors present in the baseline.

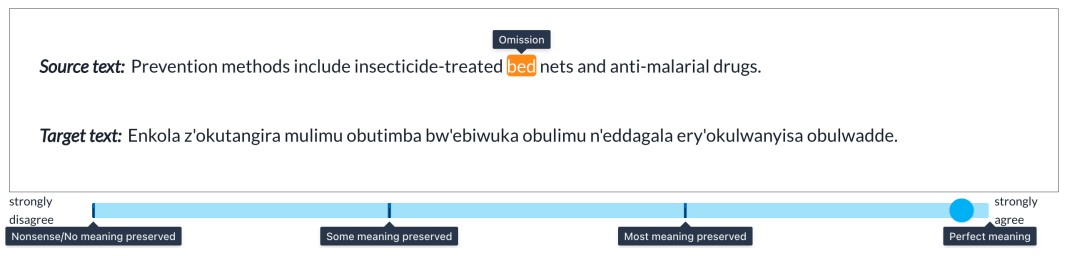

Figure 4: Evaluated translations from the MarianMT-Adapter LoRa with Active Learning.

Additionally, comparisons with widely-used translation systems such as Google Translate and the **Facebook/NLLB-200-distilled-600M** model reveal the importance of domain-specific training. Google Translate, while slightly better than some of our models, lacks the fine-tuning necessary for accurate medical translations. The **Facebook/NLLB-200-distilled-600M** model performed poorly, with a BLEU score of **27%**, highlighting the challenges of using general-purpose, distilled models in low-resource language tasks.

## 5.3 LIMITATIONS

Despite these advancements, the study is not without limitations. The reliance on BLEU scores as the primary evaluation metric may overlook nuanced translation errors, especially in a specialized domain like medical translation. BLEU, while useful for assessing overall accuracy, does not capture the contextual or cultural aspects of translation quality, which are critical for healthcare communication.

Another limitation is the availability of parallel corpora in Luganda. The lack of extensive, high-quality training data constrained the scope of improvements and hindered the ability to generalize the model beyond the malaria domain. While active learning mitigated some of these challenges by focusing on key samples, a larger dataset would likely yield even greater improvements.

## 5.4 FUTURE WORK

Future research should focus on expanding the training datasets, particularly by engaging the local community in data collection and annotation. Crowdsourcing efforts involving native speakers and medical professionals could help build a more extensive and diverse corpus for model training.

In addition, future work will incorporate more advanced active learning techniques, including different sampling strategies that go beyond uncertainty sampling. Parameter-efficient fine-tuning approaches such as prefix-tuning or prompt-tuning could also be explored to enhance model adaptability without requiring extensive retraining.

Human evaluations, conducted alongside BLEU scores, will provide a more comprehensive assessment of translation quality, particularly in capturing the nuances of medical terminology and cultural appropriateness. These human assessments will be crucial for refining models intended for healthcare applications.

## 6 CONCLUSION

This study successfully developed and optimized machine translation models for translating medical content from English to Luganda. We demonstrated that using adapter models and active learning significantly improves translation quality, particularly in handling domain-specific medical texts. The findings emphasize the importance of domain-specific data and iterative learning strategies in enhancing machine translation systems.

As we look toward future developments, expanding the available resources for low-resource languages, incorporating human evaluations, and focusing on explainability will be key steps toward improving the contextual and cultural relevance of translations. This research contributes to the broader goal of reducing language barriers in healthcare and improving access to medical information in multilingual settings.

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
