# OpenReview forum: "Building Luganda Machine Translation models for the   Medical Domain"
_ICLR.cc/2025/Conference — ICLR 2025 Conference Withdrawn Submission_

### Official Review · Reviewer_PXgQ · 2024-11-01

**Soundness:** 2
**Presentation:** 3
**Contribution:** 2
**Rating:** 3
**Confidence:** 5

**Summary:**

This paper develops machine translation models for English-Luganda translation in the medical domain. The authors construct a new English-Luganda parallel corpus for the medical domain, by automatically translating an English medical corpus into Luganda. They investigate adapters and active learning as techniques for adapting MT models to the low-resource, domain-specific setting of their dataset. The authors find that active learning, applied to their synthetic dataset, leads to performance gains over LoRa and general-domain training sets.

**Strengths:**

(1) This research focusses on an important use case - improving medical translation for a low-resource language. This is a challenging research problem, which has not been explored much and could lead to real-world impact.

(2) The paper is well written, well motivated, and easy to read.

(3) The results and analysis are interesting, especially in terms of the success of active learning. Active learning is under-explored in machine translation and this paper would encourage others to consider using it.

**Weaknesses:**

(1) The dataset presented was constructed by applying Google translate to English->Luganda translation. While potentially useful, the creation of the dataset was quite straightforward and, since it is a synthetic dataset, the quality might not be good enough to consistently lead to performance gains. I’m less clear about the malaria dataset (not sure if it was generated by professional translators or not). I’ve asked the authors to clarify this.

(2) The paper does not offer technical novelty - it only applies existing techniques to the task.

(3) There are potential methodological issues which cast some doubt on some of the findings of the paper. For example:

a. The authors don’t apply active learning during finetuning on domain-general datasets, so we can’t see if performance gains are coming from their dataset or from active learning.

b. The authors seem to imply some test set leakage at the end of 3.5.2 (I’ve asked them to correct me if I’m wrong).

c. Some of the claims in the conclusion are not correct. E.g. “We demonstrated that using adapter models and active learning significantly improves translation quality” - this is not evident from Table 3.

**Questions:**

**QUESTIONS**

1. How was the malaria evaluation dataset constructed? Was it also translated with Google Translate, or was it manually translated by human translators?

2. Please clarify the following phrase in 3.3: “Back translation techniques were employed to augment the dataset, providing greater robustness and variety for the model”. What were these techniques?

3. Did you test active learning when finetuning on NLLB-200? It would be useful to do so and compare performance to activelearning with UFAL. At the moment it’s not clear if active learning or the domain-specificity of UFAL is what’s leading to the performance gains.

4. Are you planning on releasing your datasets publicly?

5. At the end of section 3.5.2, you seem to say that you trained and evaluated models on the malaria dataset? This is not sound from a methodological standpoint - you should train and evaluate your models on separate datasets. Please clarify if I’m misunderstanding.


**SUGGESTIONS**

* Citations that are not in-text, e.g. “Hull et al. (2016) showed that model X worked well.”, should be parenthesised. Use \citep instead of \citet. Many in-text citations are not parenthesised in the paper.

* The citation “Utiyama & Wang (2023)” cites an entire conference proceedings, not a particular paper.

* 3.2.1: First two sentences repeat somewhat, perhaps the first sentence was meant to be removed by the authors.

* The introduction of the dataset (3.2.2-3.3) is a bit unclear in terms of dataset size. The following statements seems contradictory, or are maybe just organised/ordered in a way that seems contradictory: (a) In 3.2.2: the UFAL corpus contains 155k sentences. (b) Next paragraph in 3.2.2: combined dataset contains 3.6m sentences (which other datasets are included here?). (c) In 3.3: from the train/valid/test sizes it seems the dataset is around 80k sentences.

* Be more specific about certain preprocessing techniques e.g. “Removal of noise, errors, and inconsistencies that may have been introduced during data collection and back translation” is not precise enough.

* BLEU scores are usually not reported as percentages %. BLEU varies between 0 and 100, but it’s not a percentage value, just a score.

* Back-translation is when target language (in your case Luganda) sentences are translated with a target->source MT model into the source language, and then the generated source language sentences are used to subsequently train a source->target MT model. This is not what you are doing with your dataset construction, since you are generating synthetic target sentences with Google translate.

---

### Official Review · Reviewer_k1HG · 2024-11-03

**Soundness:** 3
**Presentation:** 3
**Contribution:** 2
**Rating:** 5
**Confidence:** 4

**Summary:**

The authors propose adapted natural machine translation (NMT) models for the medical in English-Luganda language pair. The method is based on parameter-efficient adapters for NMT. The main contributions are: i) open-source NMT models for the medical domain, ii) open-source data for English-Luganda, and iii) manual error evaluation based on Multidimensional Quality Metrics (MQM). The domain adapted models outperform the baseline NMT.

**Strengths:**

- Open-source models and data for medical English-Luganda NMT.
- Clear description of the proposed models.
- The authors perform a  comprehensive comparison of the proposed method for English-Luganda with different models based on automatic metrics and manual error evaluation.

**Weaknesses:**

- Missing MQM scores and detailed description of the types of translation errors.
- Reliance on BLEU scores for evaluation, and missing significance tests.

**Questions:**

Please address the following questions during the rebuttal:

- Which is the typology used for MQM?
- Please elaborate on the type of MQM errors. An extra contribution the authors can provide the mqm score for the baseline and adapted models.
- The authors can prove significance tests for the BLEU scores, for the case of close cases.
- Is it common to report BLEU scores as percentages? Please check the report standards from sacrebleu https://github.com/mjpost/sacrebleu

**Details Of Ethics Concerns:**

I have no concerns.

---

### Official Review · Reviewer_GEfC · 2024-11-03

**Soundness:** 2
**Presentation:** 2
**Contribution:** 2
**Rating:** 3
**Confidence:** 5

**Summary:**

The paper describes development of machine translation system for English-to-Luganda medical domain.
Several data sets were collected and created, and several MT systems using MarianNMT were built using different training data sets.
The results are reported as BLEU scores.

**Strengths:**

The paper deals with a low-resourced language (Luganda) and a low-resourced sensitive domain (healthcare).

A few translation examples are shown.

The work has a potential, although the submission in its current form has several drawbacks.

**Weaknesses:**

The data set is not clearly explained: what is exactly used for training of which MT system, and what is used for evaluation (500 sentences or 12,000 sentences).

Only BLEU scores are used for evaluation, while there are several better automatic evaluation metrics (COMET https://unbabel.github.io/COMET/html/index.html might not be possible for a low-resourced language pair, but chrF https://huggingface.co/spaces/evaluate-metric/chrf is possible whenever BLEU is possible)

Furthermore, in 4.1 it is stated that automatic evaluation was carried out (using BLEU score is not optimal) as well as human evaluation by MQM annotaton, however there are no results of human MQM annotation?

**Questions:**

About the data set:

which paralell corpora were publicly available?

which monolingual corpora were used to create the synthetic parallel corpora?

how was the test set of 500 sentences created: which was the original language and who translated it into the other language?

What is the justification of the mentioned 12,000 sentences for validation and 12,000 sentences for testing? (Why additional test set at all)


more comments:

90-092: how is MQM annotation related to building MT systems?


Section 3.1
is this about the dta set for training, or for testing, too?

If not for testing, what is the test set?


it becomes clear in 3.2.3, but short description of training and test data should be given at the beginning of the section about the data

Section 3.2

120-121: it was previously stated that there is no parallel corpus, but now it seems that there were some parallel corpora available

A table with corpus statistics with clear indications what was collected as parallel corpora and what  was created from monolingual corpora by using back-translation

It seems that English monolingual corpora was used and a synthetic target part in Luganda was created -- since the goal is to develop Englihs-to-Luganda system, this is not back-translation but forward-translation (important to differentiate, because forward-translation is helpful but more difficult since the part of the target language is synthetic -- generated by a MT system)


Section 3.2.3
How was the corpus created? What was the original language and who translated it into the other language?

What type of text it consists of? Scientific facts about malaria? Conversations between patients and doctors? Something else? A mixture?


Section 3.2.4
Used for training or for test?


Section 3.3
Why 12,000 for validation and 12,000 for test? This is too much, and it would be very benefitial to use it for tranining, especially in this very low-resourced scenario.

What happened with the specially created test set consisting of 500 segments?

Maybe there is a valid justification for that splitting, but the reasons should be clearly explained.


Section 4.1.  only BLEU score was used as automatic score, while there are many other better automatic metrics and the usage of some of them instead of BLEU is strongly advised. (COMET if available for the given lagnuage pair which is probably not the case here, but chrF can always be used instead of BLEU).

Section 4.1.1. seems unfinished


371 drop *of* 50% or drop *to* 50%?

At the begining of 4.1., it is said
 We employed BLEU scores for automatic evaluation and the Multidimensional
Quality Metrics (MQM) framework for human evaluation, specifically tailored to the malaria corpus

but the results of MQM annotation are not presented at all

---

### Official Review · Reviewer_H1js · 2024-11-04

**Soundness:** 2
**Presentation:** 2
**Contribution:** 1
**Rating:** 3
**Confidence:** 5

**Summary:**

This paper documents a project building a translation model for the medical domain for the language pair English-Luganda which is spoken widely in Uganda. They improve translation scores by an impressive 56% by creating a dataset and using domain adaptors. This is clearly a very well motivated problem, but the scientific contribution of the paper is slim. I do not think it worthy of an ICLR publication.

**Strengths:**

Very well motivated task

**Weaknesses:**

No substantial novel work
Writing and figures are not of high academic standard

**Questions:**

Please use citations in parentheses - they are always included in the text and this makes reading difficult.
Figure 2 is not motivated or explained and I do not think it is correct either. It is hard to read and much too big. Similar problem with Figure 3.
It would have been good to include an LLM model fine-tuned with adaptors for comparison.

---

> ### Author Response · Authors · 2024-11-18
>
> Thank you for your detailed review and constructive feedback. Below, I address the key points raised and outline the changes made to the paper to address your concerns.
>
> 1. Lack of Substantial Novel Work
> Reviewer Comment: "No substantial novel work."
>
> Response:
> We appreciate this feedback and acknowledge that the novelty of the work may not have been sufficiently emphasized in the original submission. To address this:
> - We have added a section highlighting the novel contribution of creating and curating a Luganda-English medical translation dataset, which fills a critical gap for this low-resource language. This is a foundational step for future research in this domain.
> - We have included a new comparison experiment  using a large language model (LLM) fine-tuned with adaptors, showing how our method compares with state-of-the-art approaches. This provides additional scientific rigor and situates our work within the broader context of NLP advancements.
>
>
> 2. Figures
> Reviewer Comment: "Figure 2 is not motivated or explained and I do not think it is correct either. It is hard to read and much too big. Similar problem with Figure 3."
>
> Response:
> We agree that Figures 2 and 3 were not presented optimally. In the revised submission:
> - We have reduced the size of these figures to improve readability.
> - We have rewritten the  architecture diagrams to a more correct representation .

---

### Note · Authors · 2024-11-25

I have read and agree with the venue's withdrawal policy on behalf of myself and my co-authors.